# Evaluation of Ayush-64 (a Polyherbal Formulation) and Its Ingredients in the Syrian Hamster Model for SARS-CoV-2 Infection Reveals the Preventative Potential of *Alstonia scholaris*

**DOI:** 10.3390/ph16091333

**Published:** 2023-09-21

**Authors:** Zaigham Abbas Rizvi, Upasna Madan, Manas Ranjan Tripathy, Sandeep Goswami, Shailendra Mani, Amit Awasthi, Madhu Dikshit

**Affiliations:** 1Immuno-Biology Lab, Centre for Immunobiology and Immunotherapy, Translational Health Science and Technology Institute (THSTI), NCR-Biotech Science Cluster, 3rd Milestone, Faridabad-Gurgaon Expressway, Faridabad 121001, India; upasna@thsti.res.in (U.M.); manas@thsti.res.in (M.R.T.); sandeep@thsti.res.in (S.G.); 2Immunology-Core Lab, Translational Health Science and Technology Institute (THSTI), NCR-Biotech Science Cluster, 3rd Milestone, Faridabad-Gurgaon Expressway, Faridabad 121001, India; 3Non-Communicable Disease Centre, Translational Health Science and Technology Institute (THSTI), NCR Biotech Science Cluster, 3rd Milestone, Faridabad 121001, India; shailendramani@thsti.res.in; 4Pharmacology Department, CSIR-Central Drug Research Institute, Sitapur Rd., Sector 10, Jankipuram Extension, Lucknow 226031, India

**Keywords:** Ayush-64, *Alstonia scholaris*, *Caesalpinia crista*, *Picrorhiza kurroa*, *Swertia chirata*, SARS-CoV-2, hamster model, T cell differentiation

## Abstract

In the current study, we evaluated the efficacy of Ayush-64 (A64), a polyherbal formulation containing *Alstonia scholaris* (L.) R. Br. (*A. scholaris*), *Caesalpinia crista* L. (*C. crista*), *Picrorhiza kurroa* Royle ex Benth (*P. kurroa*), and *Swertia chirata* (Roxb.) H. Karst. (*S. chirata*) against COVID-19 in a Syrian hamster infection model. Preventative use of A64 resulted in the late-phase recovery of body weight loss in severe acquired respiratory syndrome coronavirus-2 (SARS-CoV-2)-infected hamsters, suppression of pro-inflammatory cytokines, and blunted pulmonary pathology. In addition, we also investigated the efficacy of individual ingredients of A64, viz., *A. scholaris*, *C. crista*, *P. kurroa*, and *S. chirata*, in the hamster model. The hamster challenge data showed robust anti-viral and immunomodulatory potential in *A. scholaris*, followed by *P. kurroa*. However, *C. crista* and *S. chirata* of A64 showed prominent immunomodulatory potential without limiting the lung viral load. In order to better understand the immunomodulatory potential of these herbal extracts, we used an in vitro assay of helper T cell differentiation and found that *A. scholaris* mediated a more profound suppression of Th1, Th2, and Th17 cell differentiation as compared to A64 and other ingredients. Taken together, our animal study data identifies the ameliorative potential of A64 in mitigating coronavirus disease-19 (COVID-19) pulmonary pathology. *A. scholaris*, a constituent extract of A64, showed relatively higher anti-viral and immunomodulatory potential against COVID-19. The present study warrants further investigations to identify the active pharmaceutical ingredients of *A. scholaris* for further studies.

## 1. Introduction

COVID-19, caused by SARS-CoV-2, has resulted in more than 6.8 million mortalities worldwide as of 28 February 2023 (https://covid19.who.int/). The symptoms of COVID-19 range from pneumonia, fever, cough, loss of sense of smell and taste, and chest pain to dyspnea and bilateral lung infiltration in some cases [1,2,3]. The severe form of COVID-19 is characterized by acute respiratory distress (ARD), leading to hospitalization and mortality. Cytokine release syndrome (CRS) in response to acute SARS-CoV-2 infection is described as the hallmark of ARD, leading to a heightened pro-inflammatory cytokine response in the lungs [3,4,5,6]. Though an active vaccination drive has been largely successful in bringing down the morbidity and mortality arising due to COVID-19, the challenge of emerging variants with the ability to escape vaccination-induced immunity has led to the resurgent rise in COVID-19 waves across the globe [7,8]. Moreover, there is a limited benefit of vaccination in immunocompromised individuals or individuals with co-morbidity [9]. In addition, therapeutic approaches relying on anti-viral drugs such as remdesivir (RDV) and immunosuppressant dexamethasone (DXM) were shown to be successful only in a limited number of cases, either by mitigating the pathology or reducing COVID-19-related deaths [10,11]. Therefore, the search for a drug candidate with better efficacy and fewer side effects remains an area of active research that may be beneficial for current as well as future emerging coronavirus diseases.

Ayurvedic and traditional medicines, which rely on the ancient knowledge of the medicinal value of botanicals, have long been used for inflammatory and infectious diseases in many parts of Asia, including India and China [12,13,14]. In China and India, traditional herbal medicines were used both as preventatives and therapeutic regime against COVID-19. In line with this, a few randomized clinical trials were started in India and China to understand the protective efficacy. One such study from China reported 90% recovery (out of 214 patients) in COVID-19 patients with the use of traditional medicines [15,16]. Notably, studies have suggested that herbal medicines may not only improve the health status of mild to severe COVID-19 patients but also act as a preventative therapy against COVID-19 [15,16]. One such herbal formulation, A64, developed by the Central Council for Research in Ayurvedic Sciences under the Ministry of AYUSH and used for managing patients with malaria in Rajasthan and Assam states of India [17,18,19,20], was also used in India as an adjunct therapy against COVID-19. A64 is a polyherbal formulation that has previously been shown to be useful for malaria patients [18]. Moreover, recent in silico studies with the constituents of A64 indicate the possibility of inhibition of the SARS-CoV-2 main protease [21]. A64 was repurposed during the COVID-19 pandemic for the management of patients in home isolation due to its immunomodulatory and antipyretic properties [20,21]. The adjunct therapy, along with the conventional COVID-19 treatment carried out in patients with mild to moderate COVID-19 showed that the use of A64 as an adjunct therapy is safe and effective.

In the present study, we assess the efficacy of A64 and its individual herbal constituents, viz., *A. scholaris*, *C. crista*, *P. kurroa*, and *S. chirata*, in a mild model of SARS-CoV2, established in Syrian hamsters, which exhibit pulmonary pathology similar to humans [22,23]. In addition, we also evaluated their immunomodulatory potential using an in vitro T cell differentiation model [17].

*A. scholaris* is commonly called the blackboard tree or scholar tree and has been traditionally used to treat inflammatory conditions [24,25]. *C. crista* is used in traditional medicine for its anti-inflammatory and antipyretic properties [26,27]. *P. kurroa* is a perennial herb used as an Ayurvedic medicine for digestive and inflammatory conditions [28,29]. *S. chirata* belongs to the genus *Swertia*, whose members are used for various medicinal purposes, including hepatitis, inflammation, and digestive diseases, and have been previously shown to be effective against chronic fever, malaria, gastritis, etc., suggesting their potential immunomodulatory activity [30,31]. A64, a polyherbal formulation, and the mixture of the extracts of these four herbs were used for the management of early and moderate stages of COVID-19 during the pandemic due to their antipyretic and immunomodulatory properties [16,19,20,20,21,32,33].

In order to validate the efficacy of A64 against COVID-19, we used the previously established Syrian hamster model for SARS-CoV-2 infection [22,23,34]. The hamsters received preventative treatment with A64 and its individual constituents, viz., *A. scholaris*, *C. crista*, *P. kurroa*, or *S. chirata*, in human equivalent doses and were subsequently challenged with the SARS-CoV-2 ancestral strain intranasally. We next performed a cellular T cell differentiation assay to evaluate the inhibitory potential of these herbal extracts to arrest T cell differentiation and hence the effector T cell response. Taken together, we provide the first proof-of-concept study on A64 and validate the ameliorative potential of preventative A64 treatment against pulmonary pathology observed in SARS-CoV-2-infected patients.

## 2. Results

### 2.1. Effect of A64 and Its Herbal Constituents against the SARS-CoV2 Infected Hamster Model

Herbal extracts have been previously shown to exhibit immunomodulatory as well as anti-viral properties [12,13,14,35]. Here we tested the preventative efficacy of A64 and constituent herbal ingredients, *A. scholaris*, *C. crista*, *P. kurroa*, and *S. chirata*, against SARS-CoV-2 infection by using a previously established hamster model according to the scheme in Figure 1A [22,34,36,37,38]. Golden Syrian hamsters receiving A64 (I+A64) showed significant late-phase recovery in body weight when compared with infection control (I) (Figure 1B). Consistently, lungs isolated from the euthanized animals on day 4 post-challenge showed marginal amelioration in pneumonitis regions and inflammation as compared to the I group (Figure 1C). The lung viral load of the I+A64 group showed a decreasing trend as compared to the I group, but the change statistically was non-significant, suggesting that the partial body mass recovery may be due to amelioration of pulmonary pathology rather than lung viral load (Figure 1D). To understand pulmonary health in hamsters receiving A64, we evaluated the mRNA expression of lung injury markers as well as carried out histological assessments. Our lung injury marker mRNA expression data shows a significant decrease in *eotaxin* (lung injury-associated gene), *muc-1* (lung defenses against pathogenic infections), and *PAI-1* (a risk factor for thrombosis) genes in A64-treated hamsters as compared to the I group (Figure 1E). However, no significant changes in the expression of *chymase*, *tryptase* (mast cell function), and *sftp-D* (surfactant protein-D) were found in the A64 group as compared to the I group (Figure 1E). In line with mRNA expression data for lung injury markers, histological assessment of HE-stained lung sections showed amelioration in pulmonary pathology with a decrease in pneumonitis, lung injury, alveolar injury, and inflammation histological scores in the A64 group as compared to the I group (Figure 1F,G). In addition, mRNA expressions of pro-inflammatory cytokines such as *IL-4*, *IL6*, *IFN-γ*, *TNF-α*, and *IL-17A* were also found to be inhibited by preventative treatment of A64, which could be a contributing factor for mitigation of pulmonary pathology (Figure 1H). Taken together, we found that preventative treatment of A64 results in amelioration of pulmonary pathology; however, it did not reduce the overall lung viral load.

### 2.2. In Vitro Suppression of T Cell Differentiation by A64

T helper cell subsets have been shown to be an important mediator of COVID-19 immunopathology [39,40]. Since a hyper-activated effector T cell response is detrimental to pulmonary function in COVID-19, therapeutic drugs such as dexamethasone (DXM) have relied on dampening the effector T cell response and thereby reducing pulmonary pathology [41]. Effector CD4+T cells can be categorized as Th1, Th2, or Th17 cells depending on the cytokine-secretion pattern and the immunological response. Th1 cells protect against intracellular pathogens by secreting interferon (IFN)-γ, tumor necrosis factor (TNF)-β, etc. Th2 cells target parasites and secrete interleukins (IL-4, IL-5, etc.), while Th17 cells control tissue inflammation and protect against extracellular bacteria. Notably, Th1 and Th17 responses have been shown to play a role in SARS-CoV-2 infection and pathology [2,29,33,34]. In order to evaluate the immunomodulatory potential of A64 towards T cell differentiation, we studied the effect of A64 on Th1, Th2, and Th17 cell differentiation in vitro (Figure 2). A64 showed dose-dependent inhibitory potential for Th2 and Th17 differentiation but, to a lesser extent, inhibited Th1 differentiation (Figure 2A–H). Furthermore, we calculated the IC50 to evaluate A64’s inhibitory potential. A64 showed potent Th2 inhibition as compared to Th17 cell inhibition, as indicated by IC50 values (Figure 2E,H). Together, our T cell differentiation assay data shows good suppression of Th2 and Th17 cell differentiation, but not Th1 cell differentiation, by A64.

### 2.3. Anti-Viral and Immunomodulatory Potential of the A64 Ingredients

To understand the contribution of individual ingredients of A64 in the amelioration of COVID-19 pathology, we subjected each of the individual ingredients to the hamster challenge study as part of the preventative treatment regimen, as shown in Figure 3A. Hamsters receiving preventative treatment with *A. scholaris* or *C. crista* showed significant protection against body weight loss post-SARS-CoV-2 intranasal infection, while *P. kurroa* or *S. chirata* were unable to rescue them from body weight loss (Figure 3B). In line with the body mass data, excised lungs from *A. scholaris* showed lesser signs of pneumonitis and inflammation and significant inhibition of the lung viral load of infected hamsters. Moreover, *A. scholaris* showed more than 10-fold inhibition in lung viral load, followed by around 1.5-fold lung viral load inhibition by *P. kurroa* treatment (Figure 3C,D). However, we did not observe any significant inhibition of viral load in the *C. crista*-treated animals (Figure 3C,D). Since the presence of a high viral load in the lungs leads to a strong inflammatory cytokine response, pulmonary pathology, and cellular injury, we evaluated these parameters in the lungs of infected animals in the presence or absence of preventative treatment. Preventative *A. scholaris* treatment significantly reduced the cellular lung injury gene expression, followed by *P. kurroa* treatment, suggesting mitigation of pulmonary injury (Figure 3E). Histological assessment data from the hamster lungs corroborated well with the mRNA expression profile data, with significant amelioration in pulmonary pathology observed in the *A. scholaris* and *P. kurroa* groups (Figure 3F,G). In line with the pulmonary pathology data, we also found dramatic inhibition of pro-inflammatory cytokines mRNA expression, with a more than two-fold decrease in IL-6, TNF-α, and IL-17A cytokines in the *A. scholaris* and *P. kurroa* preventative groups. Cytokine release syndrome (CRS)-induced pulmonary injury is a hallmark of SARS-CoV-2 infection, and anti-inflammatory drugs that reduce the pulmonary CRS have shown clinical success as COVID-19 therapy [2,3]. It is a possibility that the anti-inflammatory properties of *A. scholaris* and *P. kurroa*, as seen by suppression of mRNA expression of pro-inflammatory cytokines, could drive the amelioration of pulmonary pathology. However, further detailed investigation of the molecular mechanism involved remains to be investigated. However, interestingly, other ingredients of A64, namely *C. crista*, *P. kurroa*, and *S. chirata*, showed some degree of anti-inflammatory potential (Figure 3H). Our data also identifies A64 ingredients such as *A. scholaris* and *P. kurroa* that have better anti-viral potential, suggesting that an improved A64 formulation with a higher percentage of *A. scholaris* and *P. kurroa* may be more effective as anti-virals. Together, our hamster study data identified *A. scholaris*, followed by *P. kurroa*, as a potent anti-viral and immunomodulatory component of A64.

### 2.4. In Vitro Effect of A. scholaris, C. crista, P. kurroa, and S. chirata on Th1 Differentiation

Th1 cells are central to the anti-viral immune response and have been associated with an effective vaccine response [42]. However, dysregulated IFN-γ secretion is also considered to be lethal to tissue health because of its cytotoxic response [43,44]. To evaluate the Th1 suppressive properties of Ayush 64 components, we evaluated the dose-response of *A. scholaris*, *C. crista*, *P. kurroa*, or *S. chirata* to inhibit the in vitro differentiation of naïve CD4+ T cells into Th1 cells. The dose-kinetic response was followed by an increasing dose from 10 µg/mL to a maximum dose of 750 µg/mL. Our intracellular flow cytometry data showed little or no inhibition of Th1 differentiation at lower doses of treatment. Moreover, even the highest treatment dose was only able to induce a 20–25% reduction in the Th1-differentiated population (Figure 4A,B). Together, all four ingredients in Ayush 64 showed a poor ability to inhibit Th1 differentiation in vitro.

### 2.5. Effect of A. scholaris, C. crista, P. kurroa, and S. chirata on Th2 Differentiation

Th2 cells are important for extracellular parasite clearance and humoral response [45]. Elevated Th2 cell cytokines have also been reported in COVID-19 cases [46]. To understand the effect of A64 ingredients on Th2 differentiation, we performed in vitro differentiation of naïve CD4+ T cells into Th2 cells. Our Th2 differentiation data showed a remarkable ability of *A. scholaris* and *C. crista* to inhibit Th2 differentiation even at lower concentrations (Figure 5A–F), while *P. kurroa* and *S. chirata* treatments showed comparably lesser inhibition of Th2 differentiation (Figure 5G–L). Notably, the IC50 for *A. scholaris* and *C. crista* for Th2 differentiation was calculated and found to be 258.8 µg/mL and 79.4 µg/mL, respectively. Taken together, we found that both *A. scholaris* and *C. crista* could inhibit Th2 differentiation in vitro, giving pharmacologically relevant IC50 values, which is suggestive of *A. scholaris* being broadly immunosuppressive against effector T cells.

### 2.6. In Vitro Effect of A. scholaris, C. crista, P. kurroa, and S. chirata Pre-Treatment on Th17 Differentiation

Th17 cells are characterized by high secretion of the IL-17A cytokine and are responsible for immunity against extracellular pathogens [47,48]. Moreover, Th17 cells are also important mediators of autoimmune and inflammatory disorders [49]. To evaluate the immunomodulatory potential of A64 ingredients against Th17 differentiation, we performed an in vitro Th17 differentiation assay in the presence or absence of treatment. Our intra-cellular flow cytometry data shows that both *A. scholaris* and *C. crista* treatment had a potent inhibitory effect on Th17 differentiation (Figure 6A–F); *P. kurroa* and *S. chirata* treatment were also effective in bringing down Th17 levels, but to a lesser extent as compared to *A. scholaris* or *C. crista* (Figure 6G–L). Subsequently, the IC50 values were found to be 347.5 µg/mL, 211.8 µg/mL, and 641.2 µg/mL, respectively, for *A. scholaris*, *C. crista*, and *S. chirata* in vitro inhibition of Th17 differentiation. Together, all four components of A64 showed inhibitory potential against Th17 differentiation in vitro.

## 3. Discussion

An active vaccination program during the pandemic has largely been successful in reducing the morbidity and mortality arising due to SARS-CoV-2 infection; however, vaccination was often compromised in offering immunity against SARS-CoV-2 variants [50,51]. Since new drug discovery is a long-term proposition, repurposing of available antiviral and immunosuppressive drugs was used extensively for the management of moderate to severely infected patients [10,11,52,53]. In addition, preventative traditional herbal medicines, including A64, were commonly used in the home setting in India during the pandemic. The availability of the hamster model for mild SARS-CoV-2 infection offered an opportunity to assess the activity of repurposed drugs and herbal medicines [13,53]. In our laboratory, we tested several herbal extracts, including A64 and its constituents, for their anti-viral and immunomodulatory potential [34,36,37,54]. In the present study, we observed that A64 did not possess profound anti-viral activity; however, it offered strong ameliorative potential against pulmonary pathology and augmented the pro-inflammatory cytokine response in the hamster model. Evaluation of the four key constituents of A64, viz., *A. scholaris*, *C. crista*, *P. kurroa*, and *S. chirata,* in the hamster model exhibited potent anti-viral and immunomodulatory activity of *A. scholaris*. *A. scholaris* also showed a profound effect on in vitro T cell differentiation. Our study thus provides scientific evidence and supports the preventative usage of Ayurvedic intervention A64 against SAR-CoV-2. It also suggests that the use of *A. scholaris* alone or in combination with *P. kurroa* could be more beneficial in preventing the viral infection.

Golden Syrian hamsters infected intranasally with SARS-CoV-2 mimic mild to moderate immuno-pathological manifestations of COVID-19 [22,23,34,55]. The hamster model has been extensively used throughout the world for screening compounds and herbal formulations for anti-viral activity and also for pulmonary pathology due to a heightened inflammatory response [13,37,56,57,58]. In the present study, we used a preventative regimen to assess its efficacy, as it has traditionally been used in many Asian countries, including India and China, to prevent the infection and also to convert the disease to a mild/moderate state. A64 is a mixture of four major herbs: *A. scholaris* (Saptaparna), *P. kurroa* (Katuki), *C. crista* (Kuberaksha), and *S. chirata* (Kiratatikta). In India, A64 has been used as a traditional medicine against mild fever, cold, and inflammation, especially during malaria, filariasis, chinkungunya, etc. [19,20]. In a clinical trial conducted in Rajasthan (n = 3600) and Assam (n = 2294) during the 1994 and 1996 malaria epidemics, respectively, it was found that preventative A64 treatment resulted in a reduced rate of infection and also lowered fever [18]. In addition, several studies have suggested the antipyretic, anti-oxidative, and anti-viral potential of its constituents [24,28,30,59]. It might have been the reason that the A64 formulation was promoted for the prevention of infection during the pandemic by the AYUSH ministry [16,17,19,20,21,33]. Indeed, studies based on in silico results have also shown that A64 could directly inhibit viral replication and may interfere with SARS-CoV-2 binding. Recently, a randomized clinical trial for COVID-19 was initiated with A64 by the Ayush ministry in India (https://iiim.res.in/cured/ayush.php accessed on 23 February 2023) [16,20,33]. However, to date, there is no scientific data based on animal studies to validate the efficacy of A64 in COVID-19. Results from the hamster challenge study demonstrated a strong anti-inflammatory potential of A64, as the mRNA expression of inflammatory cytokines such as IL-6, IL-17A, and TNF-α (all of which have been strongly correlated with COVID-19 severity) was significantly reduced. Lung pathology was also mitigated, but there was no significant reduction in the lung viral load. Notably, the mRNA expression of mast cell markers (tryptase and chymase) was not inhibited upon A64 treatment, which may be a contributing factor to disease pathology. Moreover, it is important to note that the hamster challenge model only showed mild to moderate infection, which represents the majority of the clinical cases of COVID-19. Therefore, our results might not be extrapolated to severe COVID-19 conditions and therefore need to be interpreted carefully. In addition, since hamsters lack the major reagents to evaluate deeper mechanistic insight into protective efficacy, we therefore relied on the conventional evaluation of therapeutic potential such as body mass change, lung viral load, cytokine expression, and pulmonary pathology.

A crucial finding of our study is the identification of *A. scholaris* as a potent anti-viral and immunomodulatory component of the A64 formulation. *P. kurroa* was another herbal extract that was found to decrease the pulmonary viral load. In line with this, *P. kurroa* also produced significant suppression of some of the inflammatory cytokines and cellular injury markers, with an overall suppression of the pulmonary histopathological score. Preventative treatment with *A. scholaris* showed potent protective efficacy with robust mitigation of body weight loss, lung viral load, and pulmonary pathology. An important aspect of our study is that we evaluated A64 and its constituents in the hamster COVID-19 model and in vitro assay of T-cell differentiation. We used qPCR relative mRNA expression profiling of inflammatory cytokines and cellular injury genes along with histopathological assessment to understand the factors contributing to amelioration of COVID-19. The comparative mRNA expression and histological scores are important for understanding the comparative potential of individual herbal extracts. This is critical for the holistic assessment of the herbal formulation and holds importance in defining the pharmacological potential of A64 for COVID-19 clinical cases as well as its future potential in emerging infectious diseases. In addition, our study may be implicated in the future designing of herbal formulations, which would rely on the pre-clinical efficacy data of their individual components as well as the pharmacological formulation.

Our in vivo hamster challenge study data is strongly supported by the robust T cell differentiation assay designed to evaluate the immunomodulatory potential of herbal extracts against naïve T helper cell differentiation. In vitro data showed a strong inhibitory potential for A64 as well as its four major constituents. Notably, Th1 cell differentiation was less significantly inhibited, which is desirable as Th1 cells secrete IFN-γ, a crucial mediator of the anti-viral response [60,61]. Importantly, Th2 and Th17 cells were dramatically inhibited even at lower doses of A64, *A. scholaris*, and *C. crista*. Since pro-inflammatory cytokines such as IL-4 and IL-17A are strongly correlated with COVID-19 severity and mortality, a decrease in these cytokine levels through inhibition of Th2 and Th17 differentiation seems to be beneficial in improving the health of COVID-19 patients and in preventing lung pathology.

## 4. Materials and Methods

### 4.1. Herbal Extracts

A64, or *A. scholaris*, *C. crista*, *P. kurroa*, and *S. chirata* extracts in dry powder form were provided by the National Medicinal Plant Board and were used as per pharmacopeial standards in the current study, as reported earlier [20,37], and were used for both in vitro and in vivo studies.

### 4.2. Preparation and Characterization of Extracts

Dry root powder *A. scholaris*, *C. crista*, *P. kurroa*, and *S. chirata* extracts or their herbal mixture (A64) in a ratio of 4:1 (each) were suspended in water (10 mg/mL *w*/*v*) in a shaker for 24 h at 37 °C. Thereafter, the suspension was centrifuged at 10,000× *g* for 30 min, and the supernatant was passed through a 0.45 µm filter. This filtrate (assumed to be 100% aqueous extract) is diluted in water to achieve dosing concentration. The filtrate thus obtained was used for the quality assessment of the A64 ingredients, as previously published [20].

### 4.3. SARS-CoV-2-Infected Hamster Model

Golden Syrian hamsters (mixed gender) of 6–9 weeks old and 70–100 g body weight were procured from the Central Drug Research Institute, Lucknow, and quarantined for 1 week at the small animal facility (SAF), THSTI. Thereafter, hamsters were randomly grouped (n = 5 hamsters/group to obtain statistically significant data) as uninfected (UI), infected (I), infected treated with remdesivir (I+RDV), infected with preventative A64 (I+A64) treatment, infected with preventative *A. scholaris* (I+AS) treatment, infected with preventative *C. crista* (I+CS) treatment, infected with preventative *P. kurroa* (I+PK) treatment, and infected with preventative *S. chirata* (I+SC) treatment. In order to mimic the preventative treatment regimen, which is the most common treatment regimen of traditional medicines, we used preventative treatment starting 5 days prior to infection. Each hamster from the treatment group received suspended herbal extracts (0.5% CMC preparation) twice daily (after an interval of 12 h each) as an oral dose through oral gavage. The total dose administered per day was 130 mg/kg. The dose of A64 and other ingredients was calculated on the basis of the human dose of A-64 (500 mg BD). The dosing of the herbal extract continued until the endpoint. Remdesivir, which is a prototype anti-viral drug with proven efficacy against COVID-19 in animal models, was used as the positive control for comparison [10,20,62]. The control remdesivir group received 15 mg/kg (*sc*) one day prior to and post-challenge. Intranasal infection was established with live SARS-CoV-2 (SARS-Related Coronavirus 2, Isolate USA-WA1/2020) 10^5^ PFU/100μL/hamster or with DMEM in mock control with the help of a catheter under mild injectable anesthesia (ketamine (150 mg/kg) and xylazine (10 mg/kg)) inside an Animal Biosafety Level 3 (ABSL3) facility [34,36,37]. The ancestral Wuhan strain was used for the challenge study, as most of the early described anti-viral and vaccine candidates were evaluated against the Wuhan strain [52,53]. All the protocols related to the study were approved by the Institutional Animal Ethics Committee (IAEC protocol no.: IAEC/THSTI/105), the Review Committee on Genetic Manipulation (RCGM), and the Institutional Biosafety Committee (IBSC).

### 4.4. Virus Titration

Dulbecco’s Modified Eagle Medium (DMEM) is a complete media containing 4.5 g/L D-glucose and 100,000 U/L Penicillin-Streptomycin, 100 mg/L sodium pyruvate, 25 mM (N-2-hydroxyethylpiperazine-N’-2-ethanesulfonic acid) HEPES and 2% fetal bovine serum (FBS) were used to propagate and titrate SARS-related coronavirus 2 and isolate USA-WA1/2020 virus in the Vero E6 cell line. The plaque-purified stocks of virus were prepared and used inside the (animal biosafety level 3) ABSL3 facility at the Infectious disease research facility (IDRF), THSTI, in accordance with the (institutional biosafety committee) IBSC and (review committee on genetic manipulation) RCGM protocols.

### 4.5. Gross Parameters of Infected Hamsters

Post-challenge with SARS-CoV-2, hamsters were monitored daily for their general activity and body weights were recorded. All the hamsters were sacrificed on the 4th day post-infection (dpi), which is regarded as the peak of infection, by asphyxiation in a CO_2_ chamber. This was regarded as the end-point of the study and was used to evaluate mitigation in lung viral load and pulmonary pathology [36,37,54]. No mortality was recorded before the endpoint. A necropsy was performed, and excised lung images were captured. The left lower lobe of the lungs was removed and fixed in a 10% neutral formalin solution for histological analysis [36,54]. The remaining lung samples were homogenized in Trizol for RNA isolation [34]. The homogenized samples were immediately stored at −80 °C until further use.

### 4.6. Lungs Viral Load

RNA was isolated by the Trizol-Choloform method as previously described. Quantitation of RNA yield was undertaken on a nanodrop (by measuring the concentration of DNAase-treated RNA at 260 nm), and 1 µg of RNA was reverse-transcribed to cDNA by the iScript cDNA synthesis kit (BIORAD, Hercules, CA, USA; #1708891). cDNAs were then used for qPCR by using the KAPA SYBR^®^ FAST qPCR Master Mix (5X) Universal Kit (KK4600) on the Fast 7500 Dx real-time PCR system (Applied Biosystems). qPCR results were then analyzed using SDS2.1 software [34,54]. Briefly, 200 ng of RNA was used as a template for reverse transcription-polymerase chain reaction (RT-PCR). The CDC-approved commercial kit was used for the SARS-CoV-2 N gene: 5′-GACCCCAAAATCAGCGAAAT-3′ (forward), 5′-TCTGGTTACTGCCAGTTGAATCTG-3′ (reverse). The Hypoxanthine-guanine phosphoribosyl transferase (HGPRT) gene was used as an endogenous control for normalization through quantitative RT-PCR. The relative expression of each gene was expressed as fold change and was calculated by subtracting the cycling threshold (Ct) value of hypoxanthine-guanine phosphoribosyl transferase (HGPRT-endogenous control gene) from the Ct value of the target gene (ΔCT). Fold change was then calculated according to the formula POWER (2,−ΔCT). The list of primers is provided as follows:
**Gene****Forward****Reverse***HGPRT*GATAGATCCACTCCCATAACTGTACCTTCAACAATCAAGACATTC*tryptase β2*TCGCCACTGTATCCCCTGAACTAGGCACCCTTGACTTTGC*chymase*ATGAACCACCCTCGGACACTAGAAGGGGGCTTTGCATTCC*Muc-1*CGGAAGAACTATGGGCAGCTGCCACTACTGGGTTGGTGTAAG*Sftp-D*TGAGCATGACAGACGTGGACGGCTTAGAACTCGCAGACGA*Eotaxin*ATGTGCTCTCAGGTCATCGCTCCTCAGTTGTCCCCATCCT*PAI-1*CCGTGGAACCAGAACGAGATACCAGAATGAGGCGTGTCAG*IFN-γ*TGTTGCTCTGCCTCACTCAGGAAGACGAGGTCCCCTCCATTC*TNF-α*AGAATCCGGGCAGGTCTACTTATCCCGGCAGCTTGTGTTT*IL-13*AAATGGCGGGTTCTGTGCAATATCCTCTGGGTCTTGTAGATGG*IL-17A*ATGTCCAAACACTGAGGCCAAGCGAAGTGGATCTGTTGAGGT*IL-10*GGTTGCCAAACCTTATCAGAAATGTTCACCTGTTCCACAGCCTTG*IL-6*GGACAATGACTATGTGTTGTTAGAA AGGCAAATTTCCCAATTGTATCCAG

### 4.7. Histological Assessment of the Hamster Lung Tissues

Embedded paraffin blocks prepared from fixed lung samples were sectioned and stained with HE dye as previously described. Strained lung samples were then analyzed and imaged at 40×. Histological assessment for pathological features was undertaken by a professional histologist in a blind manner, and scoring was carried out on a scale of 0–5 (where 0 indicated the absence of a histological feature while 5 indicated the highest score). The disease index score was calculated by adding all the individual histological scores.

### 4.8. In Vitro T cell Differentiation

A single-cell suspension was prepared from the spleen and lymph nodes of 6–8 weeks old C57BL/6 mice. The cells were activated using a soluble anti-CD3 monoclonal antibody (2 µg/mL) and differentiated into Th1 cells by adding recombinant mouse IL-12 (15 ng/mL) cytokine, Th2 cells by adding recombinant mouse IL-4 (15 ng/mL) cytokine, or Th17 cells by adding TGF-beta (2 ng/mL) and IL-6 cytokine (25 ng/mL). A64, *A. scholaris*, *C. crista*, *P. kurroa*, and *S. chirata* were added at the start of the culture in concentrations ranging from 10 µg/mL to 1 mg/mL [37]. After 72 h of cell culture, which is considered the optimal time point for T cell activation and cytokine release, intracellular cytokine staining was performed to check the expression of IFN-γ, IL-4, and IL-17A cytokines for Th1, Th2, and Th17 cells, respectively.

### 4.9. Intracellular Cytokine Staining

Cells were stimulated for 4 h with PMA (phorbol 12-myristate13-acetate; 50 ng/mL), ionomycin (1.0 µg/mL), and a protein-transport inhibitor containing monensin. PMA and ionomycin stimulate the immune cells in a non-antigen specific manner, and monensin is used to trap the cytokine within the cytosol. After stimulation, surface markers were stained for 15–20 min at room temperature in PBS with 1% FBS. Cells were then fixed in Cytofix and permeabilized with Perm/Wash Buffer using the BD Fixation Permeabilization solution kit and stained with anti-IL-17A (TC11-18H10.1, Biolegend, San Diego, CA, USA); anti-IFN-γ (XMG1.2, Biolegend); and anti-IL-4 (11B11, Biolegend) antibodies diluted in Perm/Wash buffer. Permeabilization was undertaken in order to make the intracellular cytokines accessible to the FACS antibodies. All antibodies were used in 1:500 dilutions. Flow cytometry was undertaken using BD FACS Symphony, and the data were analyzed with FlowJo software version 10.9.0.

### 4.10. Statistical Analysis

All the results were analyzed and plotted using GraphPad Prism 9.0 software. Body mass, viral load, and qPCR studies were compared and analyzed using one-way Analysis of Variance (ANOVA) with n = 5. *p*-value of less than 0.05 was considered statistically significant.

## 5. Conclusions

Taken together, based on our pre-clinical animal studies, we provide scientific evidence for the beneficial effect of the preventative treatment of A64, *A. scholaris*, and *P. kurroa* against COVID-19. We further demonstrate that the amelioration potential is based on the strong anti-inflammatory property, which is desirable during COVID-19. Notably, *A. scholaris* and *P. kurroa* showed a potent anti-viral effect with more than 10-fold and 1.5-fold decreases (respectively) in lung viral load. Since both *A. scholaris* and *P. kurroa* also showed strong anti-inflammatory potential along with an anti-viral response, we reason that *A. scholaris* and *P. kurroa* alone may also be an effective treatment against COVID-19. We also reason that since recent Omicron variant infections do not show aggressive pulmonary inflammation but are characterized by the presence of viral load in the lungs, *A. scholaris* and *P. kurroa* treatment may also be beneficial against these SARS-CoV-2 variants. Further studies are warranted to establish the clinical efficacy of *A. scholaris*. We reason that either *A. scholaris* or *P. kurroa* may be a better candidate for a randomized control trial than A64 for its evaluation as COVID-19 therapy. Our study also points to the need for screening and evaluation of these herbal extracts against other recurrent pathogenic infections where acute inflammatory responses are responsible for pathological manifestations.

## Figures and Tables

**Figure 1 pharmaceuticals-16-01333-f001:**
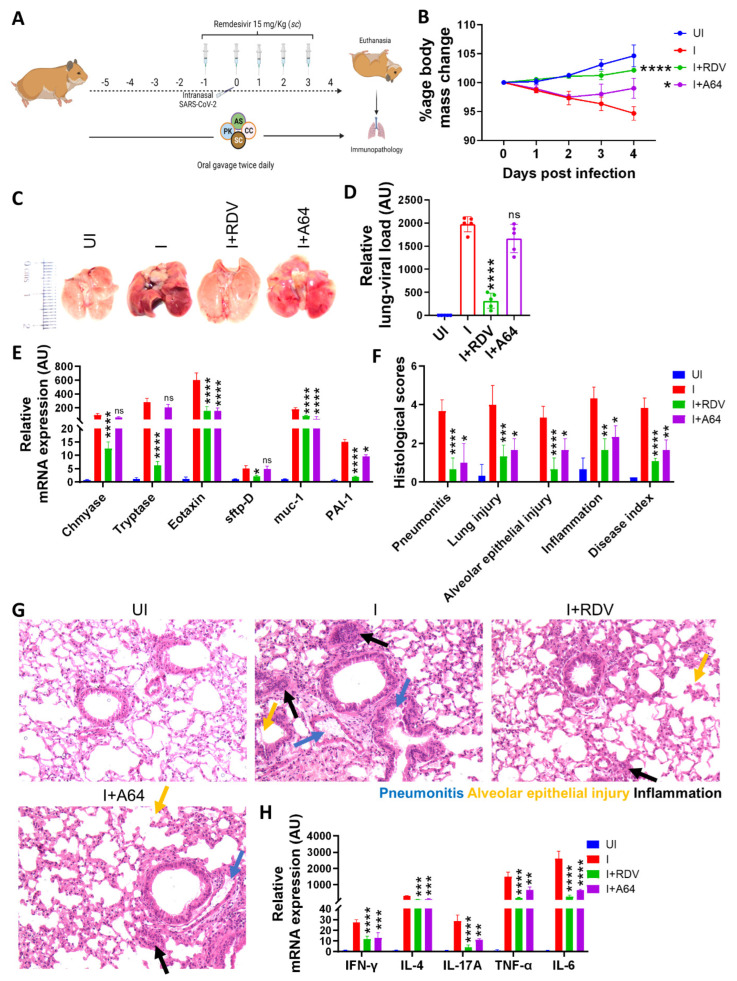
Effect of preventative treatment with A64 on SARS-CoV-2-infected hamsters. (**A**) Schematic representation of the dosing regimen for the hamster challenge study. Briefly, golden hamsters were divided into four groups: uninfected (UI), intranasally SARS-CoV-2 infected (I), infected with remdesivir treatment (I+RDV), and infected hamsters receiving preventative A64 treatment (I+A64). At 4 days post-infection (dpi), the hamsters were sacrificed, and the COVID-19 pathology was studied. (**B**) Percentage changes in the body mass of animals compared to day 0 body mass. (**C**) Representative images of hamster lungs excised after necropsy. (**D**) Relative lung viral load by N-gene expression. (**E**) Relative mRNA expression of lung injury markers. (**F**) H & E-stained lung sections were scored on a scale of 0–5 through random blinded assessment by a trained pathologist. (**G**) Representative H & E-stained lung images showing regions of pneumonitis (blue arrow), alveolar epithelial injury (yellow arrow), and inflammation (black arrow). (**H**) Relative mRNA expression of pro-inflammatory cytokines in the lungs. Bar graphs are plotted as mean + SEM. For each experiment, N = 5. One-way ANOVA using non-parametric Kruskal–Wallis test for multiple comparisons. * *p* < 0.05, ** *p* < 0.01, *** *p* < 0.001, **** *p* < 0.0001, ns = non-significant.

**Figure 2 pharmaceuticals-16-01333-f002:**
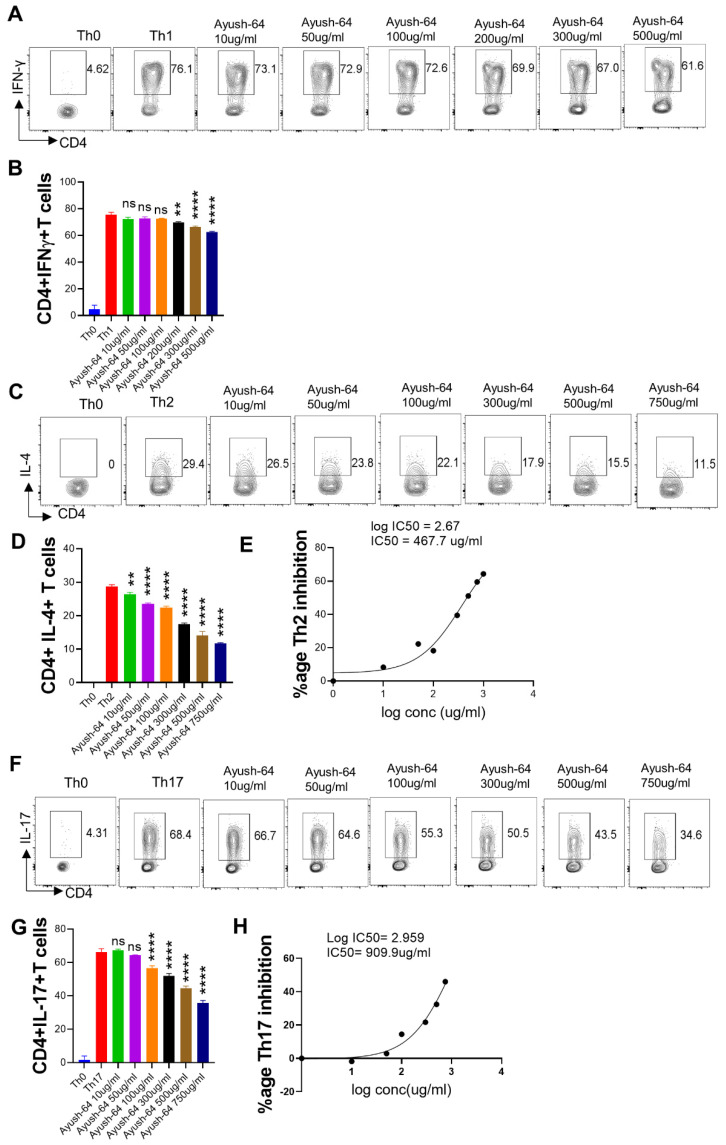
Effect of A64 on in vitro differentiation of T helper (Th1, Th2, and Th17) cells. Spleen and lymph nodes were isolated from 6–8 weeks old C57BL/6 mice, and their single-cell suspension was prepared. Cells were activated using a soluble anti-CD3 antibody and differentiated into helper Th1 (**A**,**B**), Th2 (**C**–**E**), and Th17 cells (**F**–**H**) using recombinant mouse IL-12; IL-4; and TGF-β + IL-6 cytokines, respectively. A64 was added in concentrations ranging from 10 µg/mL to 1000 µg/mL at the start of the culture. Cells were differentiated for 72 h, and IFN-γ, IL-4, and IL-17A production was measured in a CD4+ gated population by intracellular cytokine staining. IC50 values were calculated using GraphPad Prism software 9.0 (**E**,**H**). ** *p* < 0.01, **** *p* < 0.0001 by one-way ANOVA.

**Figure 3 pharmaceuticals-16-01333-f003:**
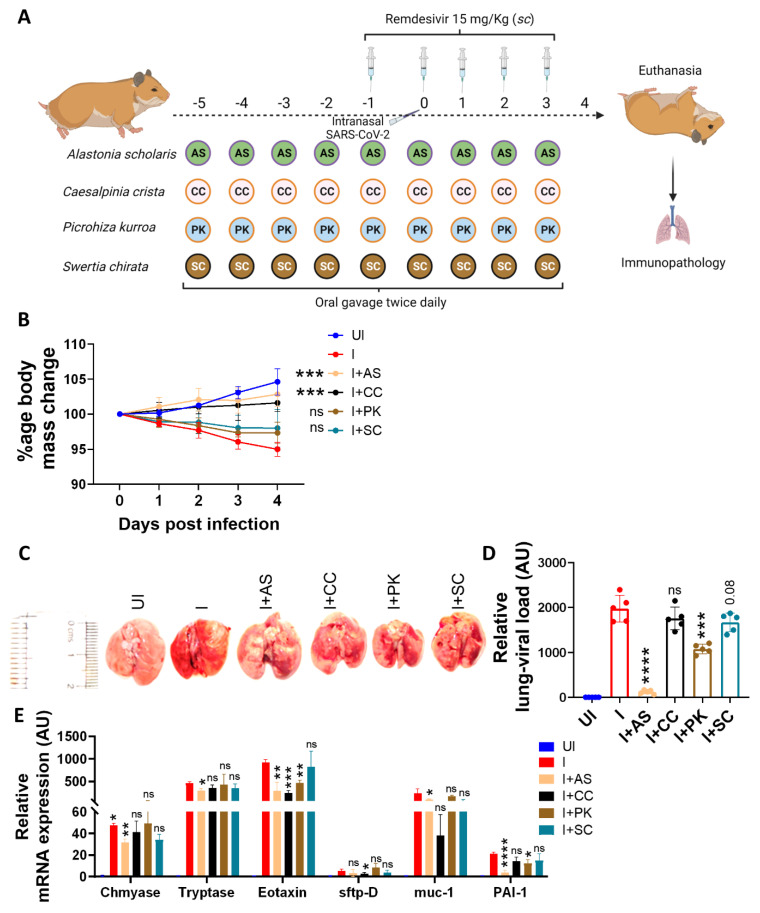
Effect of preventative treatment of *A. scholaris*, *C. crista*, *P. kurroa*, and *S. chirata* formulations on SARS-CoV-2-infected hamsters. Preventative treatment of ingredients of the A64 formulation, viz., *A. scholaris*, *C. crista*, *P. kurroa*, and *S. chirata*, was evaluated in the hamster challenge model. (**A**) Schematic representation of the study design. (**B**) Percentage changes in the body mass of animals post-challenge. (**C**) Representative images of excised lungs at 4 dpi. (**D**) Relative lung viral load by qPCR for the N-gene. (**E**) Relative mRNA expression of lung injury markers. (**F**) Representative image of H & E-stained lung sections showing regions of pneumonitis (blue arrow), alveolar epithelial injury (yellow arrow), and inflammation (black arrow). (**G**) Random blind scoring of the HE-stained lung sections was undertaken by a trained pathologist on a scale of 0–5. (**H**) Relative mRNA expression of pro-inflammatory cytokines in the lungs. Bar graphs are plotted as mean + SEM. For each experiment, N = 5. One-way ANOVA using the non-parametric Kruskal–Wallis test for multiple comparisons. * *p* < 0.05, ** *p* < 0.01, *** *p* < 0.001, **** *p* < 0.0001, ns = non-significant.

**Figure 4 pharmaceuticals-16-01333-f004:**
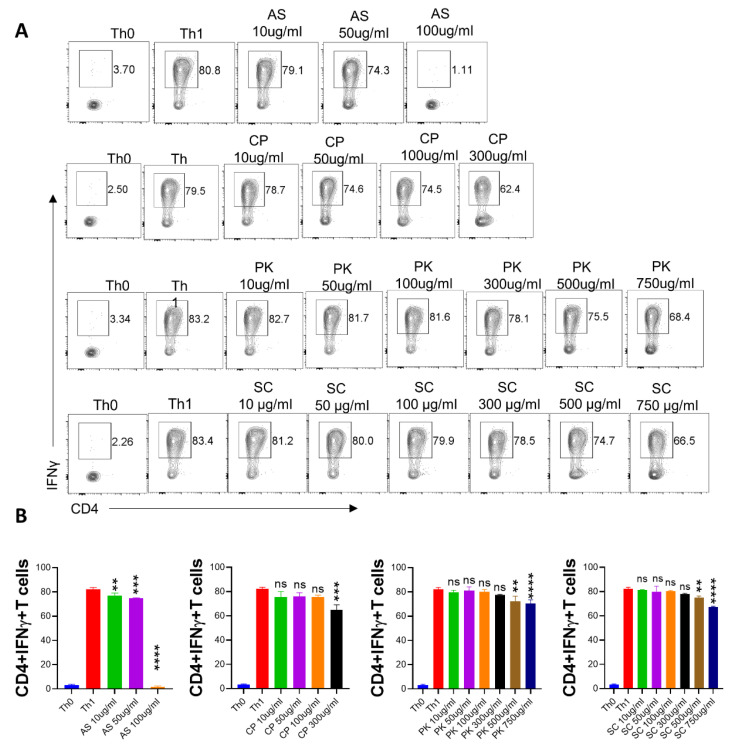
Effect of *A. scholaris*, *C. crista*, *P. kurroa*, and *S. chirata* on in vitro differentiation of Th1 cells. Spleen and lymph nodes were isolated from 6–8 weeks old C57BL/6 mice, and their single-cell suspension was prepared. Cells were activated using a soluble anti-CD3 antibody and differentiated into helper Th1 cells using recombinant mouse IL-12 cytokine. *A. scholaris*, *C. crista*, *P. kurroa*, or *S. chirata* were added in concentrations ranging from 10 µg/mL to 750 µg/mL at the start of the culture. (**A**) Cells were differentiated for 72 h, and IFN-γ production was measured by intracellular cytokine staining. (**B**) IC50 values were calculated using GraphPad Prism software 9.0. ** *p* < 0.01, *** *p* < 0.001, **** *p* < 0.0001, ns = non-significant by one-way ANOVA.

**Figure 5 pharmaceuticals-16-01333-f005:**
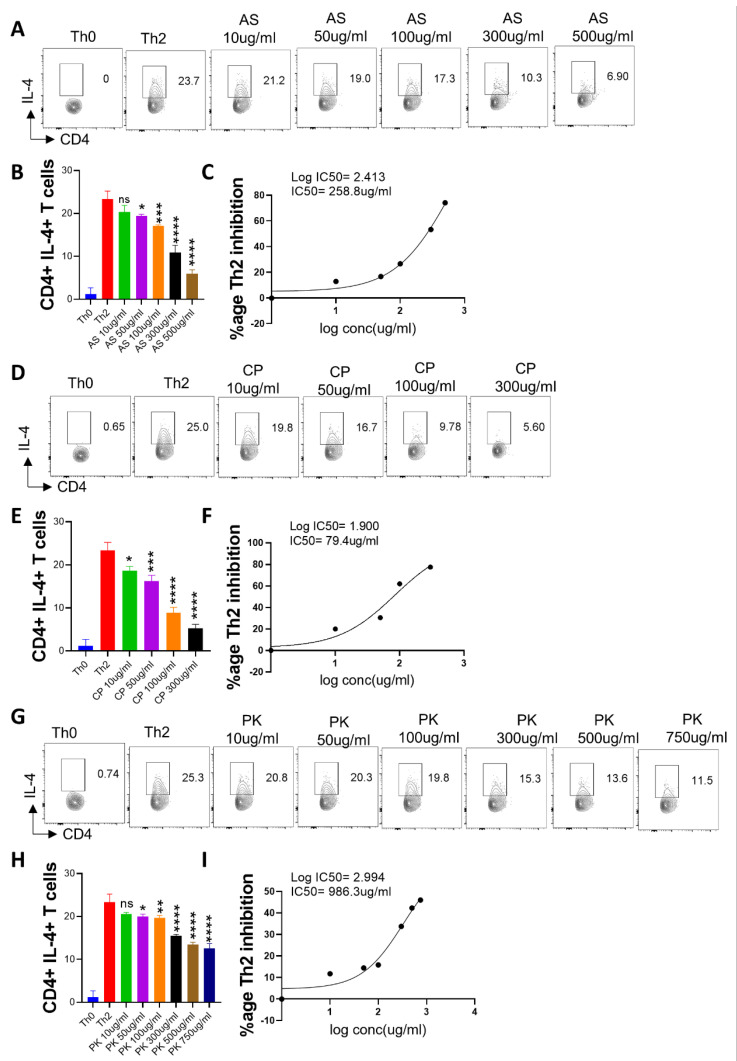
Effect of *A. scholaris*, *C. crista*, *P. kurroa*, and *S. chirata* on in vitro differentiation of Th2 cells. Spleen and lymph nodes were isolated from 6–8 weeks old C57BL/6 mice, and their single-cell suspension was prepared. Cells were activated using soluble anti-CD3 antibodies and differentiated into helper Th2 cells using recombinant mouse IL-4 cytokines. (**A**–**C**) *A. scholaris*, (**D**–**F**) *C. crista*, (**G**–**I**) *P. kurroa* , or (**J**–**L**) *S. chirata* were added in concentrations ranging from 10 µg/mL to 750 µg/mL at the start of the culture. Cells were differentiated for 72 h, and IL-4 production was measured by intracellular cytokine staining. (**B**,**C**,**E**,**F**,**H**,**I**,**K**,**L**) IC50 values were calculated using GraphPad Prism software 9.0. * *p* < 0.05, ** *p* < 0.01, *** *p* < 0.001, **** *p* < 0.0001, ns = non-significant by one-way ANOVA.

**Figure 6 pharmaceuticals-16-01333-f006:**
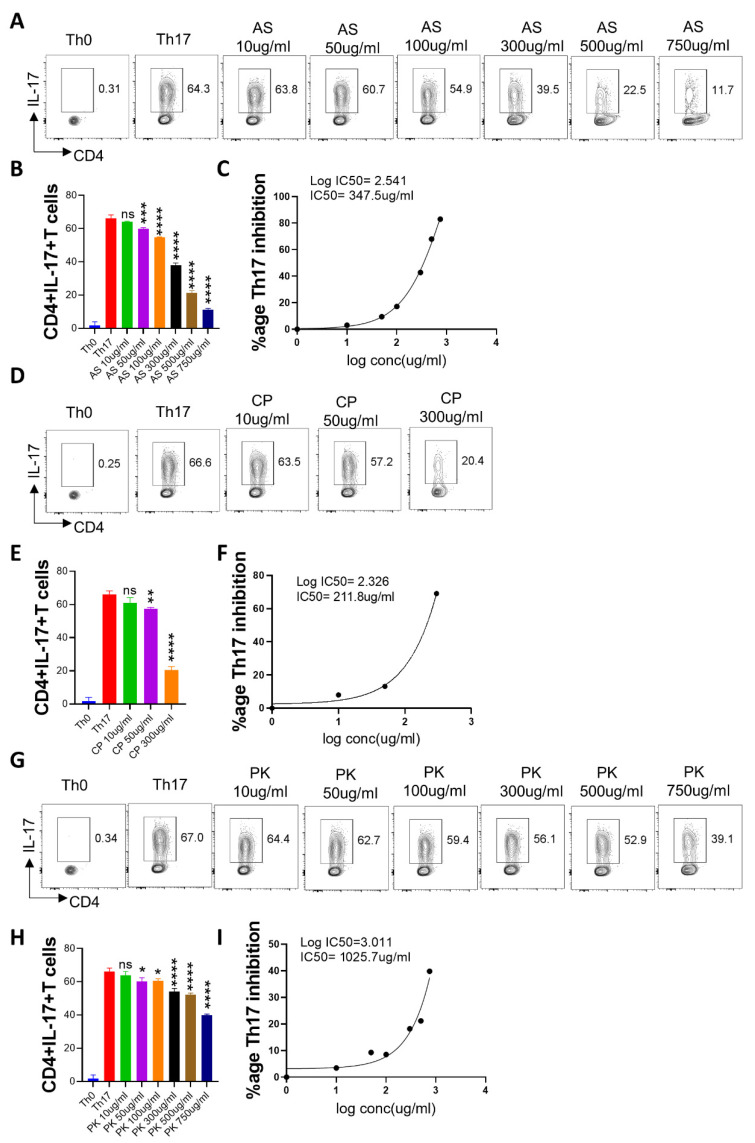
Effect of *A. scholaris*, *C. crista*, *P. kurroa*, and *S. chirata* on in vitro differentiation of Th17 cells. Spleen and lymph nodes were isolated from 6–8 weeks old C57BL/6 mice, and their single-cell suspension was prepared. Cells were activated using a soluble anti-CD3 antibody and differentiated into helper Th17 cells using recombinant mouse TGF-β + IL-6 cytokines. (**A**–**C**) *A. scholaris*, (**D**–**F**) *C. crista*, (**G**–**I**) *P. kurroa*, or (**J**–**L**) *S. chirata* were added in concentrations ranging from 10 µg/mL to 750 µg/mL at the start of the culture. Cells were differentiated for 72 h, and IL-17A production was measured by intracellular cytokine staining. (**B**,**C**,**E**,**F**,**H**,**I**,**K**,**L**) IC50 values were calculated using GraphPad Prism software 9.0. * *p* < 0.05, ** *p* < 0.01, *** *p* < 0.001, **** *p* < 0.0001, ns = non-significant by one-way ANOVA.

## Data Availability

The data that support the findings of this study are available on request from the corresponding author [A.A., M.D., Z.A.R.].

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
