# Peer review of "Evaluation of Ayush-64 (a Polyherbal Formulation) and Its Ingredients in the Syrian Hamster Model for SARS-CoV-2 Infection Reveals the Preventative Potential of *Alstonia scholaris"

_pharmaceuticals, 2023, doi:10.3390/ph16091333_

Round 1

Reviewer 1 Report

Its can be publish with minor revision

Author Response

Re: Suggestions and changes incorporated in the manuscript file in the track change mode. Thank you for helping us in improving the manuscript quality.

Reviewer 2 Report

In this manuscript, authors reported that effect of polyherbal formulation, Ayush-64, and the constituent herbal extracts on Syrian hamster model of SARS-CoV-2 and T cell differentiation. All experiments were carefully performed and well conducted to draw the conclusion. However, there are lots of problems and lacks as following:

1.      Authors mentioned the preventive effect in all contents, but in fact, the samples were administered even after viral infection. Therefore, authors should correct all “prophylactic” in the manuscript.

2.      In the lines 18 and 32 of page 1, authors should change “pre-clinical” because this is too much broad area.

3.      Please remove the author’s name from the scientific names of three plants in the Keywords section.

4.      In the Keywords, please remove (Gentianaceae) from the “Swertia chirata (Gentianaceae)”.

5.      In the lines 26, 70, and 84, viz à viz.

6.      In the line 82 of page 2, Ayush-64 à A64

7.      In the line 97 of page 2, A64, à A64; T-cell à T cell

8.      In the line 98 of page 2, what is Th cell? Please make correction or unify.

9.      In the lines 105 and 106 of page 3, extract à extracts; was à were

10.  Authors should check the sentence of lines 109-110.

11.  Authors should revise the sentence “This filterate was further used for evaluating the composition and was previously published” in the lines 113-114, page 3.

12.  Authors wrote the “SARS-Related Coronavirus 2” in the lines 126 and 134, page 3. Please explain why they wrote it this way, or they need to correct it.

13.  Authors should write the full name first for some abbreviations such as Th, ABSL3, IAEC, THSTI, RCGM, and IBSC, etc.

14.  Authors have to add the reference at “previously established hamster model” in the lines 198-199, page 5.

15.  For Figures 1B and 3B, authors should change “%age body mass change” to Body mass change (%).

16.  Authors should add marking to distinguish the graphs in Figure 1E, as shown in Figure 1F.

17.  Please use the clear ruler in Figure 1C.

18.  Please add scale bar for the Figure 1G.

19.  In the paragraph 2.8. In vitro T-cell differentiation (lines 173-180), authors should describe Th1, Th2, and Th17.

20.  Please add descriptions for Th1, Th2, and Th17 to the legend in Figure 2.

21.  In the experimental section, authors should describe the administration schedule of the samples in detail.

22.  Authors should check the administration schedule for Ayush-64 (A64) in the Figure 1A.

23.  All except the first Ayush-64 should be unified with A64.

24.  For Figure 3C, please use clear ruler.

25.  Authors should add scale bar in Figure 3F.

26.  For Figures 3G and 3H, authors should add marking to distinguish the graphs as shown in Figure 3E.

27.  For Figure 4A, please check marking especially Th1 and Th, and make correction.

28.  Please unify IL-17A, IL-17, and IL17 in the table, and in the lines 179, 186, 218, 259, 279, 340, 359, 405, and 431, and in Figures 2G, 6B, 6E, 6H, and 6K as IL-17 or IL-17A.

29.  Authors should also unify the notation for other cytokines.

30.  In Figures 5C, 5F, 5I, and 5L, please change %age Th2 inhibition to Th2 inhibition (%).

31.  In Figures 6C, 6F, 6I, and 6L, please change %age Th17 inhibition to Th17 inhibition (%).

32.  Please correct the typo on line 398, in-silicon.

33.  Authors should keep the unified journal notation with abbreviations in the reference section, especially for references 5, 15, 17, 19, 20, 25, 30, 33, 35, 36, 37, 55, 58, 59, and 64.

Since this manuscript shows many problems in English expression, authors should review the entire contents in detail. Therefore, this manuscript requires extensive editing of English language.

Author Response

  1. Authors mentioned the preventive effect in all contents, but in fact, the samples were administered even after viral infection. Therefore, authors should correct all “prophylactic” in the manuscript.

Re: We fully agree with the reviewer’s suggestion and thank them for this critical comment. We have now replaced the “prophylactic” treatment with “preventative” treatment.

  1. In lines 18 and 32 of page 1, authors should change “pre-clinical” because this is too much broad area.

Re: We thank the reviewer for their valuable comment. We have now replaced or refined the “pre-clinical” word from the manuscript.

  1. Please remove the author’s name from the scientific names of three plants in the Keywords section.

Re: Thank you. We have corrected it.

  1. In the Keywords, please remove (Gentianaceae) from the “Swertia chirata (Gentianaceae)”.

Re: Thank you. We have corrected it.

  1. In the lines 26, 70, and 84, viz à viz.

Re: Thank you. We have corrected it.

  1. In the line 82 of page 2, Ayush-64 à A64

Re: Thank you. We have now corrected it throughout the manuscript.

  1. In the line 97 of page 2, A64, à A64; T-cell à T cell

Re: Thank you. We have now corrected it throughout the manuscript.

  1. In the line 98 of page 2, what is Th cell? Please make correction or unify.

Re: Thank you. We have corrected it.

  1. In the lines 105 and 106 of page 3, extract à extracts; was à were

Re: Thank you. We have corrected it.

  1. Authors should check the sentence of lines 109-110.

Re: We thank the reviewer for their helpful comments which have helped us in improving the manuscript quality and readability.

  1. Authors should revise the sentence “This filterate was further used for evaluating the composition and was previously published” in the lines 113-114, page 3.

Re: Thank you for the valuable suggestion. We have now revised the statement to improve the clarity.

  1. Authors wrote the “SARS-Related Coronavirus 2” in the lines 126 and 134, page 3. Please explain why they wrote it this way, or they need to correct it.

Re: SARS-CoV-2 virus was initially obtained from BEI resources, CDC. The clinical isolate at BEI resources is identified by this information “NR-52281 SARS-Related Coronavirus 2, Isolate USA-WA1/2020 (Viruses)”. In order to increase the reproducibility of the results and we have written the complete source of the challenge SARS-CoV-2 strain along with its name in the manuscript.

  1. Authors should write the full name first for some abbreviations such as Th, ABSL3, IAEC, THSTI, RCGM, and IBSC, etc.

Re: Thank you for your in-depth review of the manuscript and for helping us improve the overall quality of the manuscript. We have now incorporated the full form of these above-mentioned abbreviations in the manuscript.

  1. Authors have to add the reference at “previously established hamster model” in the lines 198-199, page 5.

Re: We are thankful to the reviewer for their valuable comments. We have now added a reference to the line “previously established hamster model” 198-199, page 5.

  1. For Figures 1B and 3B, authors should change “%age body mass change” to Body mass change (%).

Re: Thank you for suggesting the changes. We have now changed the label to Body mass change (%) as suggested in Figure 1B and 3B.

  1. Authors should add marking to distinguish the graphs in Figure 1E, as shown in Figure 1F.

Re: Thank you. We have added the marking to Figure 1E.

  1. Please use the clear ruler in Figure 1C.

Re: Thank you for the valuable suggestion. We have improved the visibility of the ruler in Figure 1C.

  1. Please add scale bar for the Figure 1G.

Re: Thank you for pointing this error. We have added the scale bar in Fig 1G and Fig. 3F.

  1. In the paragraph 2.8. In vitro T-cell differentiation (lines 173-180), authors should describe Th1, Th2, and Th17.

Re: Thank you for the suggestion. We have now included the section on Th1, Th2, and Th17 cells phenotype and function details from line no. 173-179.

  1. Please add descriptions for Th1, Th2, and Th17 to the legend in Figure 2.

Re: Thanks you for the suggestions. We have revised the Figure 2 legend.

  1. In the experimental section, authors should describe the administration schedule of the samples in detail.

Re: We are thankful to the reviewer for helping us improve the manuscript quality. As suggested by the reviewer we have now added a detailed description of the dosing methodology from line no. 407-412.

  1. Authors should check the administration schedule for Ayush-64 (A64) in the Figure 1A.

Re: Thank you for pointing this out. Since A64 is a combination of AS, CC, PK, and SC we had initially represented it as AS+CC+PK+SC. However, now upon the suggestion of the reviewer, we have revised the figure and shown A64 in Figure 1A.

  1. All except the first Ayush-64 should be unified with A64.

Re: Thank you. We have corrected and unified A64 throughout the manuscript.

  1. For Figure 3C, please use clear ruler.

Re: Thank you. We have corrected it.

  1. Authors should add scale bar in Figure 3F.

Re: Thank you for pointing out this error. We have added the scale bar in Fig 1G and Fig. 3F.

  1. For Figures 3G and 3H, authors should add marking to distinguish the graphs as shown in Figure 3E.

Re: We thank the reviewer for the valuable suggestion. We have added the marking to Figures 3G and 3H.

  1. For Figure 4A, please check marking especially Th1 and Th, and make correction.

Re: Thank you. We have corrected it.

  1. Please unify IL-17A, IL-17, and IL17 in the table, and in the lines 179, 186, 218, 259, 279, 340, 359, 405, and 431, and in Figures 2G, 6B, 6E, 6H, and 6K as IL-17 or IL-17A.

Re: We thank the reviewer for their valuable input in improving the overall quality of the manuscript. We have now made it uniform as IL-17A throughout the manuscript.

  1. Authors should also unify the notation for other cytokines.

Re: Thank you for the suggestions. We have now corrected it and made it uniform throughout the manuscript.

  1. In Figures 5C, 5F, 5I, and 5L, please change %age Th2 inhibition to Th2 inhibition (%).

Re: Thank you for the suggestions. We have now corrected it and changed it to Th2 inhibition (%).

  1. In Figures 6C, 6F, 6I, and 6L, please change %age Th17 inhibition to Th17 inhibition (%).

Re: Thank you for the suggestions. We have now corrected it and changed it to Th17 (%).

  1. Please correct the typo on line 398, in-silicon.

Re: Thanks. We have corrected it.

  1. Authors should keep the unified journal notation with abbreviations in the reference section, especially for references 5, 15, 17, 19, 20, 25, 30, 33, 35, 36, 37, 55, 58, 59, and 64.

Re: Thanks. We have corrected it.

Comments on the Quality of English Language

Since this manuscript shows many problems in English expression, authors should review the entire contents in detail. Therefore, this manuscript requires extensive editing of English language.

Re: We thank the reviewer for their valuable suggestion which has helped us to greatly enhance the readability and quality of the manuscript. The English grammar and syntax have been reviewed thoroughly and corrected.

Reviewer 3 Report

The present study concerned the efficacy of the polyherbal formulation Ayush-64 (A64), against COVID-19 in an animal infection model. They evaluated the anti-viral and immunomodulatory potential of A64 together with its active pharmaceutical ingredients. The plant material was provided by National Medicinal Plant Board.

The authors obtained very promising results, especially in the case of a single component namely Alstonia scholaris (L.) R. Br (AS). 

I do hope that the authors will continue their investigations with the aim to identify the active components of AS. 

I recommend this manuscript for publication considering the current need for anti-viral preparations. 

However, I'm convinced that their research should have started with an analysis of the chemical composition of the investigated extracts.

Taking into account the kind of examined formulation (A64) which is not very common in the whole world I suggest writing more about its content. Are there any studies on its chemical content, or evaluation of its activity?

Author Response

Re: We appreciate the reviewer for the valuable comments and insightful suggestion which has helped us in improving the manuscript’s readability. We have attached a supplementary material file in which the detailed composition of Ayush64 has been described. It includes the details of the percentage of individual ingredients used for the formulation, the amount of herbal extract used, and the process of preparing the Ayush64 formulation. The QC analysis of the Ayush64 formulation was also carried out and we found very little or no contamination of microbes, heavy metals, and aflatoxins in the formulation. The formulation was further validated for QC by HPTLC and UV spectra.

In addition, in our previously published study in Frontiers pharmacology, we reported the chemical composition and characterization of Ayush64 formulation by using LC-MS/MS. A total of 11 phytoconstituents were identified putatively from AYUSH-64 extracts. In addition, in-silico studies revealed good ADME properties of most of the phytoconstituents of Ayush-64.

We have already included all of this information in the methodology and supplementary material section.

Reviewer 4 Report

The paper provides valuable insights into the potential efficacy of Ayush-64 and its herbal constituents against COVID-19. The study's design, use of pre-clinical animal models, and in vitro assays contribute to its scientific soundness. The clear presentation of results and thorough discussion of findings further enhance the paper's quality.

To improve the paper, I suggest addressing the following areas:

1. Introduction: Provide a more detailed rationale for choosing Ayush-64 and its herbal constituents for investigation. Elaborate on the context of traditional medicine in COVID-19 management and its potential as an alternative treatment option.

2. Methods: Consider providing more details on certain experimental procedures, including the rationale for specific choices, to enhance the paper's transparency and reproducibility. Also, discuss potential limitations and how they were addressed in the study.

3. Conclusions: Further elaborate on the interpretation of viral load data and the clinical relevance of the findings. Discuss the implications of the study's results for human patients and potential safety considerations.

Consider discussing the potential impact on statistical power and generalizability of the results.

4. Mechanistic Insights: While the study highlights the effects of Ayush-64 and its constituents, deeper mechanistic insights could strengthen the paper. Consider discussing the underlying cellular and molecular pathways that contribute to the observed effects.

5. Generalization of Findings: Address the generalizability of the study's findings beyond the hamster model and discuss the need for further validation in other animal models or human clinical trials.

Incorporating these suggestions would enhance the paper's overall impact, clarify its contributions, and provide a more comprehensive understanding of the potential efficacy of Ayush-64 and its herbal constituents in COVID-19 management. The paper's valuable insights can contribute significantly to the field of COVID-19 research and the exploration of traditional medicine as a complementary approach in disease management.

Author Response

  1. Introduction: Provide a more detailed rationale for choosing Ayush-64 and its herbal constituents for investigation. Elaborate on the context of traditional medicine in COVID-19 management and its potential as an alternative treatment option.

Re: Thank you for providing valuable input. As suggested by the reviewer, we have now incorporated the rationale behind choosing Ayush64 and background on the use of traditional medicine in COVID-19 management in the Introduction section from lines no 60-79 in the track change mode.

  1. Methods: Consider providing more details on certain experimental procedures, including the rationale for specific choices, to enhance the paper's transparency and reproducibility. Also, discuss potential limitations and how they were addressed in the study.

Re: We thank the reviewer for the insightful suggestions which were valuable for improving the manuscript quality. In line with the suggestions made by the reviewer, we have incorporated a more detailed methodology and have added more information related to the experiments on line no. 380, 385-396, 411, 448, 452-455, 460-461.

  1. Conclusions: Further elaborate on the interpretation of viral load data and the clinical relevance of the findings. Discuss the implications of the study's results for human patients and potential safety considerations.

Consider discussing the potential impact on statistical power and generalizability of the results.

Re: We are grateful to the reviewer for the valuable insight and constructive comments. We have now improved the Conclusion section of the manuscript discussing more on the clinical implications of the results from line no. 472-483.

  1. Mechanistic Insights: While the study highlights the effects of Ayush-64 and its constituents, deeper mechanistic insights could strengthen the paper. Consider discussing the underlying cellular and molecular pathways that contribute to the observed effects.

Re: We thank the reviewer for the insightful comments and valuable suggestions. In the manuscript, we described the amelioration potential of Ayush64 and its constituents against COVID-19 by using a hamster model for SARS-CoV-2 infection. We also used the lung samples to evaluate the mRNA expression profile of some of the cellular injury genes and cytokine expression which are known to influence the inflammation in the lungs and cause pulmonary pathology (Line 334-336 & 355-360). Since there are very less commercially available antibodies against hamsters, we used an in-vitro T cell differentiation assay to understand the immunomodulatory potential of these herbal extracts. We have now provided a more detailed discussion of these cellular and molecular pathways that may contribute to the observed effects in the discussion section. However, owing to the limitation in the biological reagents against hamsters, we do not have more data on the pathways involved. The molecular and cellular mechanisms could be investigated in greater detail by using hACE2 transgenic mice as a separate study which has the advantage of having bio-reagents to carry out molecular and cellular investigations.

  1. Generalization of Findings: Address the generalizability of the study's findings beyond the hamster model and discuss the need for further validation in other animal models or human clinical trials.

Re: Thank you for your valuable input. We have now incorporated information of the generalization of findings in the conclusion section (Line no. 389-392).

Incorporating these suggestions would enhance the paper's overall impact, clarify its contributions, and provide a more comprehensive understanding of the potential efficacy of Ayush-64 and its herbal constituents in COVID-19 management. The paper's valuable insights can contribute significantly to the field of COVID-19 research and the exploration of traditional medicine as a complementary approach in disease management.

Re: We thank the reviewer for the valuable suggestions in helping us improve the manuscript quality.

Reviewer 5 Report

The research article investigates the efficacy of Ayush-64 (A64), a polyherbal formulation containing Alstonia scholaris (AS), Caesalpinia crista (CC), Picrorhiza kurroa (PK), and Swertia chirata (SC), against COVID-19 in a Syrian hamster infection model. The study evaluates the prophylactic use of A64 and its individual herbal constituents in terms of body weight recovery, suppression of pro-inflammatory cytokines, and pulmonary pathology. In-vitro assays of helper T-cell differentiation are used to assess the immunomodulatory potential of the herbal extracts. The results suggest that A64 exhibits anti-inflammatory potential without significant reduction in the lung viral load. AS shows higher anti-viral and immunomodulatory potential compared to other herbal extracts. The study highlights the potential of A64 in mitigating COVID-19 pulmonary pathology and warrants further investigation to identify active pharmaceutical ingredients.

Major Revision:

1.     The title suffers from reflecting the final results implying reorganizing the title highlighting the AS not the polyherbal formulation which fails to reduce viral load.

2.     The abstract needs to be harmonized:

a.         Sometimes you are saying ingredient, sometimes individual extract. Make uniform.

b.         Since the PK was next to AS, why is it excluded from the further recommendation?

3.     The introduction section has failed to opt for enough clarity while a larger portion of this section encompasses the results which are unwanted. Additionally, the authors have illustrated the justified background information for all other polyherbal constituents except SC.

4.     Even though the authors received the sample powder from an organization, they need to incorporate the identification accession number the organization preserves.

5.     How was the dose 130 mg/kg bw fixed for prophylactic treatment?

6.     No toxicity of the sample was tested which could be supportive to fix the dose. However, the toxicity of such a formulation is inevitable.

7.     In section 2.6, how did the authors quantify RNA using nanodrop? What principle was adopted to quantify?

8.     The use of abbreviation should be started with using the full form (elaboration) at least once.

9.     Line 220-221; Taken together, we found that prophylactic treatment of A64 results in amelioration of pulmonary pathology however, it did not reduce the overall lung viral load-how does it reflect the overall results?

10.  The section 3.3.; its out of the study goal. What do you mean by the "in order to determine the active ingredient of A64 formulation"? You did not identify the active ingredient indeed, right?

11.  Line 268-289; The treatment of AS or CC showed significant protection in body weight loss post SARS-CoV-2; while AS showed less signs of pneumonitis and inflammation and significant inhibition of the lung viral load of infected hamster; however nothing is said about CC. What was the impact of CC?

12.  PK was unable to rescue the body weight loss but the mRNA expression looks ameliorating the pulmonary pathology--------------where is the analogical discussion? Pathology covers a broad spectrum, what did you mean by this?

13.  In section 3.1, you said viral load was not reduced, is it not contradictory with the previous section? Well, the previous one was Ayush and now it is AS. But effect of constituent should be reflected through the formulation, what do you think? Why did the formulation not reduce viral load while AS was found to do that?

14.  Line 350, what do you mean by some level?

15.  Line 373-380; in light of these statements, the title of the manuscript needs to be revised.

16.  Line 398, what do you mean by in-silicon?

17.  Line 406; Lung pathology was also mitigated, but there was no significant reduction in the lung viral load-if so, how was the involvement of virus for lung pathology ensured?

18.  The discussion part seems very poor to reflect the results through critical analysis, presenting deeper insights and analogical compatibility.

19.  208-210: Suggested to perform the statistical analyses once again.

20.  3.3: Missing in methods and material section. Model should be discussed thoroughly including the dose selection process. Safety and toxicity analysis should be included.

Minor Revision:

1.     Texual/word formatting is demanded to be revised rigorously (pls see the comments adhered to the PDF file).

2.     Ununiform organization of letters in the same word.

3.     Abbreviation approach needs to be harmonized.

4.     Line 365: Reference needed.

5.     Check for typographical errors and punctuation mistakes.

6.     There are lot more comments in the PDF file, need to be addressed.

Particularly the formatting, texualizing, using of standard approach for units need to improved. Comments enclosed in the PDF file

Author Response

The research article investigates the efficacy of Ayush-64 (A64), a polyherbal formulation containing Alstonia scholaris (AS), Caesalpinia crista (CC), Picrorhiza kurroa (PK), and Swertia chirata (SC), against COVID-19 in a Syrian hamster infection model. The study evaluates the prophylactic use of A64 and its individual herbal constituents in terms of body weight recovery, suppression of pro-inflammatory cytokines, and pulmonary pathology. In-vitro assays of helper T-cell differentiation are used to assess the immunomodulatory potential of the herbal extracts. The results suggest that A64 exhibits anti-inflammatory potential without significant reduction in the lung viral load. AS shows higher anti-viral and immunomodulatory potential compared to other herbal extracts. The study highlights the potential of A64 in mitigating COVID-19 pulmonary pathology and warrants further investigation to identify active pharmaceutical ingredients.

Major Revision:

  1. The title suffers from reflecting the final results implying reorganizing the title highlighting the AS not the poly-herbal formulation which fails to reduce viral load.

Re: We thank the reviewer for their valuable suggestion. We have revised the manuscript title with a focus on Alstonia scholaris.

  1. The abstract needs to be harmonized:
  2. Sometimes you are saying ingredient, sometimes individual extract. Make uniform.

Re: We are grateful to the reviewer for their valuable comments which has helped us in improving the manuscript quality. We have removed ‘individual extract’ and changed it with ‘ingredient’.

  1. Since the PK was next to AS, why is it excluded from the further recommendation?

Re: Thank you for a very valid observation that we missed discussing in the manuscript. The PK-treated group indeed showed a 1.5-2-folds decrease in lung viral load and the potential to suppress pro-inflammatory cytokine mRNA expression level. We have now discussed the results of PK in more detail in the methodology and discussion section wherever possible. The detail of PK results is included on pg. no. 188-205, 334-348, 482-495.

  1. The introduction section has failed to opt for enough clarity while a larger portion of this section encompasses the results which are unwanted. Additionally, the authors have illustrated the justified background information for all other poly-herbal constituents except SC.

Re: Thank you for the valuable suggestion. We have now improved the introduction section by incorporating a more detailed rationale for using herbal extracts for COVID-19 and also Ayush-64 (lines no 60-79). As suggested we have also improved the rationale for using SC in the introduction section (lines no. 89-92).

  1. Even though the authors received the sample powder from an organization, they need to incorporate the identification accession number the organization preserves.

Re:  Ayush-64 contained aqueous extracts (100 mg each) of Alstonia scholaris (bark), Picrorhiza kurroa (rhizome), Swertia chirata (whole plant), and Caesalpinia crista (200 mg seed powder). AYUSH 64 was procured by NMPB from Indian Medicines Pharmaceutical Corporation Limited (IMPCL), Uttarakhand. The manufacturing facility was a certified ISO 9001 facility (2008) and followed good manufacturing practices’ guidelines in the Ayurvedic Pharmacopoeia of India. Details of composition, quality standards, and features of chemistry, manufacturing, and controls are described in (Arvind Chopra, Girish Tillu, Kuldeep Chuadhary, Govind Reddy, Alok Srivastava, Muffazal Lakdawala, Dilip Gode, Himanshu Reddy, Sanjay Tamboli, Manjit Saluja, Sanjeev Sarmukaddam, Manohar Gundeti, Ashwini Kumar Raut, B. C. S. Rao, Babita Yadav, Narayanam Srikanth, Bhushan Patwardhan. Co-administration of AYUSH 64 as an adjunct to standard of care in mild and moderate COVID-19: A randomized, controlled, multicentric clinical trial. PLoS ONE 18(3): e0282688. https://doi.org/ 10.1371/journal.pone.0282688).

Other relevant references are mentioned below.

Narayanam Srikanth, Adarsh Kumar, Bhogavalli Chandrasekharao, Richa Singhal, Babita Yadav, Shruti Khanduri, Sophia Jameela, Amit Kumar Rai, Arunabh Tripathi, Rakesh Rana, Azeem Ahmad, Bhagwan Sahai Sharma, Ankit Jaiswal, Rajesh Kotecha. Disease characteristics, Care-Seeking Behavior, and Outcomes Associated with the Use of AYUSH-64 in COVID-19 Patients in Home Isolation in India: A Community-Based Cross-Sectional Analysis. Frontiers in Public Health (2022)

Amit Kumar Rai, Azeem Ahmad, Pallavi Mundada, Krishna Kumar V, Babita Yadav, Shruti Khanduri, Bhogavalli Chandrasekhararao, and Narayanam Srikanth. Efficacy and safety of AYUSH‑64 as standalone or adjunct to standard care in COVID‑19: a structured summary of the protocol for a systematic review. Systematic Reviews volume 11, Article number: 103 (2022)

Ashok Kumar Panda, Sarbeswar Kar, Amit Kumar Rai, B.C.S. Rao, N. Srikanth. AYUSH- 64: A potential therapeutic agent in COVID-19. Journal of Ayurveda and Integrative Medicine

Volume 13, Issue 2, April–June 2022, 100538

  1. How was the dose of 130 mg/kg bw fixed for prophylactic treatment?

Re: This was calculated on the basis of the human dose of A-64 (500mg BD)

  1. No toxicity of the sample was tested which could be supportive to fix the dose. However, the toxicity of such a formulation is inevitable.

Re: A-64 is already in human use and during the COVID-19 pandemic it was already being used for preventive purposes.

  1. In section 2.6, how did the authors quantify RNA using nanodrop? What principle was adopted to quantify?

Re: We thank the reviewer for their valuable comment. We have included the principle adopted for RNA quantitation at line no. 432-433.

  1. The use of abbreviation should be started with using the full form (elaboration) at least once.

Re: Thanks. We have noted it and corrected it everywhere to the best of our understanding.

  1. Line 220-221; Taken together, we found that prophylactic treatment of A64 results in amelioration of pulmonary pathology however, it did not reduce the overall lung viral load-how does it reflect the overall results?

Re: We appreciate this comment from the reviewer which is indeed a very interesting finding coming out from our study. We found though A64 was not able to significantly reduce the lung viral load it does produce a significant amelioration of pulmonary pathology especially reduced histological score and mRNA expression of lung injury markers. Notably, we also observed a significant reduction in the mRNA expression of inflammatory cytokines upon A64 treatment. Cytokine release syndrome (CRS) induced pulmonary injury is a hallmark of SARS-CoV-2 infection and anti-inflammatory drugs that reduce the pulmonary CRS have shown clinical success as COVID-19 therapy (15). It is a possibility that the anti-inflammatory properties of A64, as seen by the suppression of mRNA expression of pro-inflammatory cytokines, could drive the amelioration of pulmonary pathology. However, further detailed investigation of the molecular mechanism involved remains to be investigated.

  1. J. B. Moore, C. H. June, Cytokine release syndrome in severe COVID-19. Science. 368, 473–474 (2020).
  2. J.-B. Lascarrou, COVID-19-related ARDS: one disease, two trajectories, and several unanswered questions. The Lancet Respiratory Medicine. 9, 1345–1347 (2021).
  3. D. M. Del Valle, S. Kim-Schulze, H.-H. Huang, N. D. Beckmann, S. Nirenberg, B. Wang, Y. Lavin, T. H. Swartz, D. Madduri, A. Stock, T. U. Marron, H. Xie, M. Patel, K. Tuballes, O. Van Oekelen, A. Rahman, P. Kovatch, J. A. Aberg, E. Schadt, S. Jagannath, M. Mazumdar, A. W. Charney, A. Firpo-Betancourt, D. R. Mendu, J. Jhang, D. Reich, K. Sigel, C. Cordon-Cardo, M. Feldmann, S. Parekh, M. Merad, S. Gnjatic, An inflammatory cytokine signature predicts COVID-19 severity and survival. Nat Med. 26, 1636–1643 (2020).
  4. A. K. Panda, S. Kar, A. K. Rai, B. C. S. Rao, N. Srikanth, AYUSH- 64: A potential therapeutic agent in COVID-19. Journal of Ayurveda and Integrative Medicine. 13, 100538 (2022).
  5. RECOVERY Collaborative Group, P. Horby, W. S. Lim, J. R. Emberson, M. Mafham, J. L. Bell, L. Linsell, N. Staplin, C. Brightling, A. Ustianowski, E. Elmahi, B. Prudon, C. Green, T. Felton, D. Chadwick, K. Rege, C. Fegan, L. C. Chappell, S. N. Faust, T. Jaki, K. Jeffery, A. Montgomery, K. Rowan, E. Juszczak, J. K. Baillie, R. Haynes, M. J. Landray, Dexamethasone in Hospitalized Patients with Covid-19. N Engl J Med. 384, 693–704 (2021).
  6. The section 3.3.; it’s out of the study goal. What do you mean by the "in order to determine the active ingredient of A64 formulation"? You did not identify the active ingredient indeed, right?

Re: We regret the ambiguity in the statement and we express our gratitude to the reviewer for the insightful comment which has helped us in improving the overall quality and readability of the manuscript. We have now revised the statement and corrected it.

  1. Line 268-289; The treatment of AS or CC showed significant protection in body weight loss post SARS-CoV-2; while AS showed less signs of pneumonitis and inflammation and significant inhibition of the lung viral load of infected hamster; however, nothing is said about CC. What was the impact of CC?

Re: Thank you for that important suggestion. We have now revised the result section and have discussed about the lung viral load of CC on line no. 192-193.

  1. PK was unable to rescue the body weight loss but the mRNA expression looks ameliorating the pulmonary pathology--------------where is the analogical discussion? Pathology covers a broad spectrum, what did you mean by this?

Re: We thank the reviewer for this insightful comment. We have now described the results of PK in a greater detail in line no. 207-214.

  1. In section 3.1, you said viral load was not reduced, is it not contradictory with the previous section? Well, the previous one was Ayush and now it is AS. But effect of constituent should be reflected through the formulation, what do you think? Why did the formulation not reduce viral load while AS was found to do that?

Re: We thank the reviewer for their valuable suggestions. We have now revised the result section and have discussed the observations in order to make it more coherent to the readers. We thank the reviewer for improving the overall readability of the manuscript.

  1. Line 350, what do you mean by some level?

Re: Thank you. We have revised the statement in the manuscript to remove the ambiguity.

  1. Line 373-380; in light of these statements, the title of the manuscript needs to be revised.

Re: We thank the reviewer for their valuable suggestion. We have revised the manuscript title with the focus on A. scholaris.

  1. Line 398, what do you mean by in-silicon?

Re: We regret for this typo. We have corrected it to in-silico.

  1. Line 406; Lung pathology was also mitigated, but there was no significant reduction in the lung viral load-if so, how was the involvement of virus for lung pathology ensured?

Re: Thank you for asking this important question. We have now included a discussion on it in the discussion section.

  1. The discussion part seems very poor to reflect the results through critical analysis, presenting deeper insights and analogical compatibility.

Re: We thank the reviewer for their valuable comments and suggestions and helping us in improving the manuscript quality and readability. We have now substantially revised the discussion section with new points.

  1. 208-210: Suggested to perform the statistical analyses once again.

Re: Thanks. We affirm that the statistical analysis was checked and no changes are required in it.

  1. 3.3: Missing in methods and material section. Model should be discussed thoroughly including the dose selection process. Safety and toxicity analysis should be included.

Re: The selection of AYUSH-64 was on the basis of Ayurvedic logic and clinical experience. It was also found useful to treat malaria, cough and other mild respiratory tract infections and other disorders. A comprehensive description of AYUSH 64 with a therapeutic potential in COVID-19 has been published

[Ahmad S, Zahiruddin S, Parveen B, Basist P, Parveen A Gaurav, et al. Indian Medicinal Plants and Formulations and Their Potential Against COVID-19-Preclinical and Clinical Research. Front Pharmacol. 2020; 11: 578970. https://doi.org/10.3389/fphar.2020.578970 PMID: 33737875].

Other relevant references are listed below

Amit Kumar Rai, Azeem Ahmad, Pallavi Mundada, Krishna Kumar V, Babita Yadav, Shruti Khanduri, Bhogavalli Chandrasekhararao, and Narayanam Srikanth. Efficacy and safety of AYUSH‑64 as standalone or adjunct to standard care in COVID‑19: a structured summary of the protocol for a systematic review. Systematic Reviews volume 11, Article number: 103 (2022)

Minor Revision:

  1. Texual/word formatting is demanded to be revised rigorously (pls see the comments adhered to the PDF file).

Re: Thank you. We have revised the MS file rigorously and have addressed the comments and concern raised by the reviewer.

  1. Ununiform organization of letters in the same word.

Re: Thank you. We have made the abbreviation, words, etc uniform throughout the manuscript. We are grateful to the careful revision of the manuscript.

  1. Abbreviation approach needs to be harmonized.

Re: Thank you. We have improved it.

  1. Line 365: Reference needed.

Re: Thank you. We have included the reference now.

  1. Check for typographical errors and punctuation mistakes.

Re: Thank you. We have thoroughly revised the manuscript by checking the typographical errors and punctuation mistakes.

  1. There are lot more comments in the PDF file, need to be addressed.

Re: We are grateful to the reviewer for their in-depth review and helping us in improving the overall quality of the manuscript. We have addressed all the comments of the PDF file and also incorporated a rebuttal for the same in the pdf file. Also, we have also incorporated the suggested changes in the track change mode in the manuscript.

Round 2

Reviewer 2 Report

I read carefully this revised manuscript entitled “Evaluation of Ayush-64 (a polyherbal formulation) and it’s ingredients in Syrian hamster model for SARS-CoV-2 infection reveals preventative potential of A. scholaris”. Authors responded all in the cover letter for the comments I had pointed out. However, revised version had corrections for some of them. In particular, authors missed to respond to the comment on the figures, and also omitted reference part in the revised manuscript. And additional comments are as follows:

1.      In the title, authors should use a full scientific name for “A. scholaris”.

2.      For short title, please change “Protective efficacy study” to protective effect.

3.      In the Keywords, authors should replace the “and” between Picrorhiza kurroa and Swertia chirata with “;”.

Minor editing of English language required

Author Response

Comments and Suggestions for Authors

I read carefully this revised manuscript entitled “Evaluation of Ayush-64 (a polyherbal formulation) and it’s ingredients in Syrian hamster model for SARS-CoV-2 infection reveals preventative potential of A. scholaris”. Authors responded all in the cover letter for the comments I had pointed out. However, revised version had corrections for some of them. In particular, authors missed to respond to the comment on the figures, and also omitted reference part in the revised manuscript. And additional comments are as follows:

Re: We are grateful to the reviewers for their valuable comments and insightful suggestions. We have now tried to address all the comments raised by reviewers to the best of our knowledge.

  1. In the title, authors should use a full scientific name for “ scholaris”.

Re: We thank the reviewer for their valuable inputs and helping us improve the manuscript quality. We have now corrected the title as suggested.

  1. For short title, please change “Protective efficacy study” to protective effect.

Re: Thank you. We have corrected the short title.

  1. In the Keywords, authors should replace the “and” between Picrorhiza kurroaand Swertia chirata with “;”.

Re: Thank you. We have corrected the keywords.

Comments on the Quality of English Language

Minor editing of English language required

Re: We thank the reviewer for their valuable insights in improving the manuscript quality and readability. We have now gone through the manuscript and have addressed the grammatical and syntax errors.

Reviewer 5 Report

Authors did not address the comments embedded in the PDF file. I suggest to recheck the comments and address the comments regarding Figures 

Enclosed in the comments

Author Response

Reviewer 5

Comments and Suggestions for Authors

Authors did not address the comments embedded in the PDF file. I suggest to recheck the comments and address the comments regarding Figures 

Re: We are grateful to the reviewers for their valuable comments and insightful suggestions. We have now tried to address all the pdf embedded comments raised by reviewers to the best of our knowledge.

Comments on the Quality of English Language

Re: We thank the reviewer for their valuable insights in improving the manuscript quality and readability. We have now gone through the manuscript and have addressed the grammatical and syntax errors.

Round 3

Reviewer 2 Report

I read carefully this second revised manuscript. However, the revisions according to comment 33 still cannot be verified because authors again omitted the reference part in the second revised manuscript. I recommend this work to be published in the ‘pharmaceuticals’ after checking the reference section.